# Stress-Tolerant Yeasts: Opportunistic Pathogenicity Versus Biocontrol Potential

**DOI:** 10.3390/genes10010042

**Published:** 2019-01-14

**Authors:** Janja Zajc, Cene Gostinčar, Anja Černoša, Nina Gunde-Cimerman

**Affiliations:** 1Department of Biotechnology and Systems Biology, National Institute of Biology, Večna pot 111, SI-1000 Ljubljana, Slovenia; janja.zajc@nib.si (J.Z.); anja.cernosa@gmail.com (A.Č.); 2Department of Biology, Biotechnical Faculty, University of Ljubljana, Jamnikarjeva 101, SI-1000 Ljubljana, Slovenia; cene.gostincar@bf.uni-lj.si; 3Institut ‘Jožef Stefan’, Jamova cesta 39, SI-1000 Ljubljana, Slovenia

**Keywords:** opportunistic pathogen, biocontrol agent, virulence, stress tolerance, secretome, CAZy, protease, thermotolerance, oligotrophism, melanin, siderophore, biofilm

## Abstract

Stress-tolerant fungi that can thrive under various environmental extremes are highly desirable for their application to biological control, as an alternative to chemicals for pest management. However, in fungi, the mechanisms of stress tolerance might also have roles in mammal opportunism. We tested five species with high biocontrol potential in agriculture (*Aureobasidium pullulans*, *Debayomyces hansenii*, *Meyerozyma guilliermondii*, *Metschnikowia fructicola*, *Rhodotorula mucilaginosa*) and two species recognized as emerging opportunistic human pathogens (*Exophiala dermatitidis*, *Aureobasidium melanogenum*) for growth under oligotrophic conditions and at 37 °C, and for tolerance to oxidative stress, formation of biofilms, production of hydrolytic enzymes and siderophores, and use of hydrocarbons as sole carbon source. The results show large overlap between traits desirable for biocontrol and traits linked to opportunism (growth under oligotrophic conditions, production of siderophores, high oxidative stress tolerance, and specific enzyme activities). Based on existing knowledge and these data, we suggest that oligotrophism and thermotolerance together with siderophore production at 37 °C, urease activity, melanization, and biofilm production are the main traits that increase the potential for fungi to cause opportunistic infections in mammals. These traits should be carefully considered when assessing safety of potential biocontrol agents.

## 1. Introduction

Biocontrol agents employ various and interlinking mechanisms to inhibit or outcompete growth of pathogenic microorganisms such as antibiosis by secretion of numerous compounds (e.g., siderophores, antibiotics, toxins, etc.); direct parasitism by being armored with various lytic enzymes [1] and finally by efficient competition for nutrients and space. Besides, substantial tolerance to variable stress factors and efficient competition for limiting nutrients are highly desirable traits for robust and efficient biocontrol agents used against plant pathogens when these agents are administered both preharvest and postharvest [2]. On the other hand, a certain degree of extremotolerance, enabling the establishment of the infection, is also observed in fungi that are associated with opportunistic infections in humans and other mammals [3]. Indeed, significant co-occurrence of extremotolerance (e.g., osmotolerance and psychrotolerance) and opportunism was shown at the level of fungal orders in a kingdom-wide phylogenetic analysis [4].

Here, we studied tolerance to various stress factors, biofilm formation and growth on different sources of carbon of seven stress-tolerant species of yeasts, among which five have been suggested as potential biocontrol agents in agriculture (i.e., *Aureobasidium pullulans*, *Debayomyces hansenii*, *Meyerozyma guilliermondii*, *Metchnikowia fructicola* and *Rhodotorula mucilaginosa)* and two are well-known opportunistic pathogens (i.e., *Exophiala dermatitidis* and *Aureobasidium melanogenum*).

Among these species, *A. pullulans* and *A. melanogenum* (order, Dothideales; class, Dothideomycetes; subphylum, Pezizomycotina; phylum, Ascomycota) are a good example of how closely related species in the same genus can show very different behaviors, particularly considering that these two species were known as varieties of *A. pullulans* until 2014 [5]. *A. pullulans* is considered to be safe and is used in the biocontrol of plant pathogens, while *A. melanogenum* is associated with human opportunistic infections. Due to the recent redefinition of the *Aureobasidium* species [5], it is in most cases impossible to distinguish between *A. pullulans* and *A. melanogenum* in the previous literature, therefore the knowledge of their lifestyles and other traits is often ambiguous. *A. melanogenum* is found mainly in oligotrophic aqueous environments (including tap water) [6], while *A. pullulans* is more frequently encountered in mildly osmotic environments, and is associated with the phylosphere and carposphere of various plants (reviewed in [5]). *A. pullulans* has considerable biotechnological potential as a producer of hydrolytic enzymes and antimicrobial compounds [7] and it is commercially available as a biocontrol agent against preharvest and postharvest diseases of fruits and vegetables (e.g., products such as Blossom Protect^®^, Boni Protect^®^, AureoGold^®^). *A. melanogenum* on the other hand is recognized as an emerging human opportunist [8] that is responsible for several infections in immunocompromised patients, ranging from cutaneous, ocular, catheter-related, pulmonary and peritonitis infections, to systemic (reviewed in [5]). Both species are polyextremotolerant–able to survive hypersaline (up to 3.0 M NaCl) [9], acidic and basic (pH 3–10) [10], cold (4 °C) [11] and oligotrophic [12] conditions to a certain degree.

The melanized yeast *E. dermatitidis* (Chaetothyriales, Eurotiomycetes, Pezizomycotina, Ascomycota) has been found in several anthropogenic habitats, including household dishwashers [13,14], humidifiers [15], saunas and steam baths [16,17,18], sink drains and water taps [19,20], and environments that are polluted with hydrocarbons, such as creosote-treated and concrete petroleum-oil-contaminated railway sleepers [21,22]. Surprisingly, despite its pronounced thermotolerance, *E. dermatitidis* has also been isolated from glacial ice [23]. *E. dermatitidis* is truly polyextremotolerant as it is halotolerant (up to 2.0 M NaCl), acidotolerant (pH 2) and alkalitolerant (pH 10) and both psychrotolerant and thermotolerant (able to grow from 0 °C to 47 °C) [13,24,25]. It is a well-recognized opportunistic human pathogen that is showing increasing incidence of infections [26], which can be superficial, cutaneous or subcutaneous and visceral or systemic, with sometimes fatal outcomes. Indeed, the documented infections related to *E. dermatitidis* now include (but are not limited to): onychomycosis, keratitis, otitis externa, cutaneous and subcutaneous phaeohyphomycosis, endocarditis, liver cirrhosis, pneumonia, and brain infection (reviewed in [3]).

*Rhodotorula mucilaginosa* (*incertae sedis*, Pucciniomycotina, Basidiomycota), is a basidiomycetous yeast with characteristically pink/red-colored colonies, and it has been isolated from numerous cold [27] and hypersaline [28] environments (similar to those for *A. pullulans*), and from poplars and other plant surfaces [29,30], effluent of a uranium mineral heap [31], copper-contaminated wastewater sediment [32], nitrobenzene-contaminated active sludge [33], acidic water [34], dairy plant surfaces after disinfection [35] and oligotrophic Patagonian lakes [36,37]. It is acidotolerant (pH 2.2) [3] and oligotrophic [31] and can grow over a broad temperature range (from 0.5–5 °C to 37 °C) [3]. *R. mucilaginosa* is psychrotolerant [38], as well as highly heat resistant (some cultures survived 10 min exposure to 62.5 °C), thus frequently causing spoiling of heat-treated and refrigerated food [3,30]. On the one hand it shows promising results when used as a biocontrol agent for fruits stored at low temperatures [39,40,41], but on the other hand it is considered to be an emerging opportunistic pathogen that can cause fungemia associated with catheters, endocarditis, peritonitis, meningitis, and endophthalmitis [42].

*Meyerozyma guilliermondii* (Debaryomycetaceae, Saccharomycetales, Saccharomycotina, Ascomycota), previously known as *Pichia guilliermondii*, is another yeast that is of interest due to its potential use in biological control against spoilage fungi and in biotechnological applications, as well as for its clinical importance (reviewed in [43]). *M. guilliermondii* is widely distributed in environments such as oil-containing soils, plant leaves and vegetables, lake and sea water, insects, processed foods (reviewed in [43]), dishwashers [13,44], washing machines [20] and tap water [6]. It can also be a part of the human skin and mucosa microbiome [45] and is a causative agent of various clinical manifestations such as skin lesions, cellulitis and osteomyelitis albeit with low incidence even in immunocompromised patients [45]. It is well adapted to changing environmental conditions, although its most apparent trait is its thermotolerance (able to grow at 40 °C) [46,47].

*Debaryomyces hansenii* (Debaryomycetaceae, Saccharomycetales, Saccharomycotina, Ascomycota) has a synonymous anamorph known as *Candida famata*, and it is often encountered in hypersaline natural environments and salted food. *D. hansenii* plays a role in the ripening processes of some types of cheese [48]. Its most obvious characteristic is its halotolerance (up to 2.0 M NaCl) [49], which has been studied extensively [50,51]. *D. hansenii* has been suggested for biological control of postharvest rots in citrus fruit [52]. It is not a commonly encountered human opportunist, although it has been reported to be responsible for various types of infections [53,54].

*Metschnikowia fructicola* (Metschnikowiaceae, Saccharomycetales, Saccharomycotina, Ascomycota) has been reported almost exclusively from the carposphere of apples [55] and grapes [56] and it is an efficient biological control agent of postharvest diseases of fruit and vegetables, as the commercially available product “Shemer^TM^” [57]. *M. fructicola* is acidotolerant (pH 2.0) [58] and moderately resistant to heat shock and oxidative stress [59]. There are no reports on its pathogenic potential, and it shows no toxicity, infectivity, or pathogenicity according to European Food Safety Authority tests.

Of the above-presented species, several strains were chosen to assess the traits that might define their roles in biocontrol or/and in virulence, such as growth under oligotrophic conditions and at human body temperature, biofilm formation, tolerance to oxidative stress, production of siderophores and hydrolytic enzymes, and use of different hydrocarbons as sole carbon source. As polyextremotolerance is both a desirable characteristic for biocontrol agents and is linked to virulence in mammals, an overlap of traits was expected between these representatives of biocontrol groups of species and opportunists. These traits that evolved as adaptations to environmental stress and that can serve as preadaptation (exaptation) to facilitate colonization of animal hosts were investigated as markers of their biocontrol potential and the safety of their use.

## 2. Materials and Methods

### 2.1. Yeast Strains and Culture Conditions

The selected yeast strains (Table 1) are preserved at the Culture Collection Ex Infrastructural Centre Mycosmo (accession numbers noted by EXF), (Department of Biology, Biotechnical Faculty, University of Ljubljana, Ljubljana, Slovenia), and the Westerdijk Fungal Biodiversity Institute (Utrecht, The Netherlands) (accession numbers noted by CBS). The cultures were maintained on defined yeast nitrogen base (YNB) medium (pH 7.0) composed of 0.17% (*w*/*v*) yeast nitrogen base (Qbiogene, Carlsbad, CA, USA), 0.5% ammonium sulphate (Sigma-Aldrich, Saint Louis, MO, USA), 2% glucose (Kemika, Ovada, Italy) and 2% agar (Formedium, Hunstanton, United Kingdom), in deionized water.

### 2.2. Growth under Oligotrophic Conditions and at 37 °C

To test the growth of the strains under oligotrophic conditions and at human body temperature (37 °C), cell suspensions were prepared from cultures grown on YNB medium (pH 7.0) in deionized water to OD_600_ = 0.5. Five microliters of cell suspension was spotted on YNB, diluted YNB medium (0.01×) and agar medium (2% agar; Formedium) and incubated at 24 °C and 37 °C for 14 days.

### 2.3. Biofilm Quantification

Biofilm formation by the strains was quantified using the crystal violet assay, as previously described [60], with modifications. Briefly, cultures under the exponential growth were adjusted to OD_590_ = 0.15 in 0.1× YNB. One hundred microliters of cell suspension was inoculated into Nunc-TSP plates (Thermo Scientific, Waltham, MA, USA) as four parallel samples. After 48 h of stationary incubation at 24 °C 100 μL 1% (*w*/*v*) crystal violet was added to each well. After 30 min the wells were washed with deionized water and 100 μL 10% (*w*/*v*) sodium dodecyl sulphate was added to solubilize the biofilm. The absorbance of the resulting suspension was measured at 590 nm (A_590_; CytationI3 Imaging reader) using the Gen5 Microplate Reader and Imager Software (both BioTek Instruments, Inc, Winooski, VT, USA). Each experiment was performed twice, and the data were analyzed using the PAST 3.20 software [61].

### 2.4. Enzymatic Activities

The enzymatic activities of selected strains were determined on agar medium (i.e., amylase, cellulose, xylanase, pectinase, β-glucosidase, chitinase, esterase, urease) or in tubes (i.e., gelatinase and keratinase). Cultures grown on YNB medium (pH 7.0) were washed in deionized water, adjusted to OD_600_ = 1.0, and used for inoculations by spotting 5 µL cell suspension onto the agar surface, or as 20 µL cell suspensions in test tubes. All the assays were incubated at 24 °C. When possible (i.e., amylase, cellulase, xylanase, pectinase) the enzymatic index (EI) was determined across three replicates, according to Equation (1):EI = (diameter of colony and precipitation/clearing zone)(diameter of the colony)^−1^(1)

The amylolytic activity was determined on starch agar, following a modified procedure of Dodman and Reinke (1982) [62]. The medium was 1% soluble starch, 0.2% NaNO_3_, 0.05% KCl, 0.001% MgSO_4_ × 7H_2_O, 0.1% KH_2_PO_4_, 0.1% zinc solution, 0.1% copper solution (all Sigma-Aldrich) and 1.2% agar in deionized water. After incubation for 14 days the plates were exposed to iodine vapors (Sigma-Aldrich) in a closed container. The amylolytic activity of each strain was detected as a clear zone around the colonies.

The cellulolytic activity was determined using carboxymethyl cellulose (CMC; Sigma-Aldrich) medium, which was prepared according to Paterson and Bridge (1994) [63]. The CMC medium (pH 5.2) was composed of 0.1% NH_4_H_2_PO_4_, 0.02% KCl, 0.02% MgSO_4_ × 7H_2_O, 0.02% CaCl_2_ (all Sigma-Aldrich), as 4% CMC and 1.2% agar in deionized water. Similarly, a xylan medium (pH 4.2) was prepared, where 4% xylan was used instead 4% CMC. After 8 days of incubation the CMC and xylan plates were stained with 0.3% Congo Red (Merck, Darmstadt, Germany) solution for 15 to 30 min, and washed with 1 M NaCl (Fisher Scientific, Hampton, NH, USA). A clear zone around the colonies was considered as positive reaction, which was measured.

The pectinase activity was determined according to previously published procedures [64,65]. Briefly, the medium (pH 7.0) used was composed of 0.05% KH_2_PO_4_, 0.01% MgSO_4_ × 7H_2_O, 0.02% NaCl, 0.02% CaCl_2_ × 2H_2_O, 0.001% FeCl_3_ × 6H_2_0, 0.5% apple pectin (Sigma-Aldrich) as 1.5% agar in deionized water. After 14 days of incubation the plates were flooded with iodine solution and a positive reaction was defined as a clear zone around the colonies.

The β-glucosidase activity was determined on aesculin agar, which was composed of 0.2% NaNO_3_, 0.05% KCl, 0.001% MgSO_4_ × 7H_2_O, 0.1% KH_2_PO_4_, 0.1% zinc solution, 0.1% copper solution, 0.5% sucrose, 0.5% aesculin (Sigma-Aldrich), 0.02% ferric citrate, as 2% agar in deionized water. A positive reaction was defined as a black complex in the medium, which was due to the reaction between ferric citrate and the degradation product of aesculin (aesculetin) [63].

The chitinase activity was determined on a medium that contained colloidal chitin that was prepared as follows: 5 g chitin (from crab shells; Sigma-Aldrich) was ground with a pestle and mortar in the presence of liquid nitrogen until fine powder was obtained, then chitin powder was treated with 100 mL 12 M HCl in a 500 mL beaker with continuous stirring carried out in a chemical fume hood at 30 °C. This was diluted with 900 mL deionized water, and incubated overnight at 4 °C. The chitin–HCl mixture was then passed through a cellulose filter using a Buchner filtration system. The filtrate was washed five times with ice-cold deionized water by centrifugation at 8000× *g*, for 10 min at 4 °C. Then the colloidal chitin was neutralized with 1 M KOH and washed three times. Finally, the colloidal chitin was resuspended in deionized water and added to the base medium to a final 0.5% (*w*/*v*). The medium (pH 4.7) for detection of the chitinase activity according to Agrawal and Kotasthane (2012) was 0.03% MgSO_4_ × 7H_2_O, 0.3% (NH_4_)_2_SO_4_, 0.2% KH_2_PO_4_, 0.1% citric acid monohydrate, 0.5% colloidal chitin, 0.015% bromocresol purple (Sigma-Aldrich), 0.2% Tween-80 (Sigma-Aldrich) as 1.5% agar in deionized water. Chitin degradation was scored to be positive if there was a purple zone around the colonies [66].

The esterase activity was determined according to Lelliott and Stead (1987) [67], using the esterase medium (pH 5.4) composed of 1% Tween-80, 1% Peptone (Conda Pronadisa, Torrejon de Ardoz, Spain), 0.5% NaCl, 0.01% CaCl_2_ × 2H_2_O, 0.0025% Bromocresol purple as 1.5% agar in deionized water. The esterase activity was considered positive if the plate became purple with a white zone of precipitate around the colonies [67].

The proteolytic activity was determined on casein, prepared according to Brizzio et al. (2007) [68]. This was defined as a clear zone around the colonies [68,69].

The gelatinase activity was determined in 5 mL nutrient gelatine tubes, composed of 12% gelatine (Sigma-Aldrich), 0.3% beef extract (Becton Dickinson, Franklin Lakes, NJ, USA) and 0.5% peptone (Merck). After the growth of yeasts, the tubes were incubated for 30 min at 4 °C and then examined for liquefaction of the gelatine, which was interpreted as a positive reaction [70].

The keratinase activity was determined in tubes containing 2 mL agar basal medium without keratin, and then overlaid with 1 mL alkaline (pH 9) keratin azure (4 mg/mL) medium. After inoculation and incubation, the tubes were examined for the release and diffusion of the azure dye into the lower layer of the basal medium [71].

The urease activity was determined using a urea-containing medium, which was prepared as follows: first, the basic medium was prepared as 0.1% NH_4_H_2_PO_4_, 0.02% KCl, 0.02% MgSO_4_ × 7H_2_O, 0.005% bromcresol purple, 1% glucose as 1.2% agar in deionized water. An aqueous solution of 50% (*w*/*v*) urea was sterilized by filtration and added to the autoclaved basic medium to the final concentration of 2% urea. To evaluate the urease activity, both control (without urea) and urea-containing test media were inoculated. Only the strains where the control plates remained yellow (no pH increase) and the test plates turned from red to purple (increased pH due to urea hydrolysis) were considered as positive [63].

### 2.5. Tolerance to H_2_O_2_ Oxidative Stress

To determine the tolerance to H_2_O_2_ oxidative stress, the cell suspensions of the cultures were prepared in liquid YNB medium at OD_600_ = 0.5. The cell suspensions were treated with 0 mM H_2_O_2_ (control conditions), 10 mM H_2_O_2_ and 20 mM H_2_O_2_ (Carlo Erba, Milano, Italy) for 30 min and 1 h at 24 °C and with shaking at 300 rpm. After the treatment, the cell suspensions were serially diluted (10^−1^, 10^−2^, 10^−3^, 10^−4^) in deionized water with 5 µL of each dilution then spotted onto YNB agar plates and incubated at 24 °C for 7 days.

### 2.6. Screening of Siderophore Production

Production of siderophores was determined using the chrome azurol S (CAS) agar plate assay, as previously described [72,73]. Briefly, two solutions were prepared: first as 10 mL of 1 mM FeCl_3_ × 6H_2_O in 10 mM HCl, and mixed with 50 mL CAS solution and 40 mL hexadecyltrimethylammonium bromide; and the second as 30.24 g PIPES (all Sigma-Aldrich), 12 g of the 50 (*w*/*v*) NaOH, 20 g malt extract, 1 g peptone, 20 g glucose and 20 g agar in 900 mL deionized water. Both solutions were autoclaved separately and combined when they had cooled. The agar plates were centrally inoculated with 5 µL cell suspension in deionized water with OD_600_ = 1.0 and incubated at 24 °C and 37 °C.

### 2.7. Assimilation of Hydrocarbons

The assimilation of hydrocarbons was determined according to Satow et al. (2008) [74]. Briefly, liquid YNB (pH 7.0) without any carbon source was used as the basic growth medium. Three different culture conditions were used: (i) YNB medium as control; (ii) YNB supplemented with 20% (*w*/*v*) of mineral oil (Sigma-Aldrich) and (iii) YNB supplemented with 20% (*w*/*v*) of *n*-hexadecane (Sigma-Aldrich) as sole carbon source. The mineral oil and *n*-hexadecane were filter-sterilized. Tubes were inoculated with 100 µL cell suspension in deionized water at OD_600_ = 1.0 and incubated statically at 24 °C. Assimilation of the cyclic hydrocarbon toluene was determined according to Prenafeta Boldú et al. (2001) [75] on plates with YNB and purified agar medium (both lacking carbon source; pH 7.0) under toluene vapors. These were inoculated with 10 µL cell suspension at OD_600_ = 1.0. The plates were incubated at 24 °C in a desiccator under a toluene atmosphere: 3% (*v*/*v*) toluene (Merck) solution in dibutyl phthalate (Sigma-Aldrich). Growth was determined by comparison of the agar plates incubated without and with toluene vapors [74,75].

### 2.8. In Silico Analysis

The in silico analysis used the predicted proteome sequences of *A. pullulans* (EXF-150) [5], *A. melanogenum* (CBS 110374) [5], *D. hansenii* (CBS 767) [76], *M. fructicola* 277 [77], *R. mucilaginosa* (ATCC 58901) [78], *M. guilliermondii* (ATCC 6260) [79] and *E. dermatitidis* (UT8656) [80] retrieved from MycoCosm (The fungal genomics resource; Joint Genome Institute).

The numbers of secreted proteins were estimated by screening the entire proteomes using the SignalP 4.1 software [81] and then subtracting from the identified sets of proteins all those that contained a trans membrane domain, as identified by the TMHMM v2.0c software [82]. Carbohydrate active enzymes (CAZy) were identified in the resulting collection of proteins using the dbCAN server (http://cys.bios.niu.edu/dbCAN2/). Proteins were considered as CAZy if they were identified as such by at least two of the three tools used (i.e., HMMER, DIAMOND, Hotpep). Visualization of the CAZy numbers was performed with the corrplot package in R [83].

Non-ribosomal peptide synthetases (NRPSs) were identified by analysis of the whole proteomes with a stand-alone version of the antiSMASH software, using the default parameters [84]. The adenylation (A) domains were also determined by using antiSMASH, and identified by inclusion in the phylogenetic analyses (performed as described below for enzymes involved in oxidative stress responses) together with the adenylation domains from previously characterized NRPSs from *A. nidulans*, *A. fumigatus* and *C. heterostrophus* [85,86,87,88]. While one NRPS can contain several adenylation domains (with all included in the phylogenetic analysis), the number of proteins counted as NRPSs in each species are reported regardless of the number of adenylation domains they contained.

Identification of the genes that encode catalases, superoxide dismutases (SODs) and peroxiredoxins was performed as described in Gostinčar et al. (2014) [89]. Briefly, the Kyoto Encyclopedia of Genes and Genomes (KEGG) proteins from a selection of fungi were used as queries in a blastp search of the predicted proteomes investigated here, with an e-value threshold 10^−11^. The phylogenetic tree was constructed for each group of putative orthologues, by alignment of the sequences with the MAFFT 7.215 software [90], estimation of the best protein evolution model and alpha parameter of the gamma distribution of the substitution rate categories with the ProtTest 3.4.2 software and estimation of the phylogenies with PhyML 3.3 software [91], with aLRT implementation for calculation of the branch supports as Chi^2^-based supports. Using the inferred phylogenies, all the proteins with unexpected phylogenetic positions or large phylogenetic distances from other proteins in the tree were searched on a case-by-case basis against the non-redundant GenBank protein database, and their putative functions were either confirmed or they were removed from the dataset. Visualization of the CAZy numbers was performed using the corrplot package in R [83].

Urease (accession number, XP_572365) [92] and genes of the toluene degradation pathway [93] were used for the blastp search of the predicted fungal proteomes with e-value cut-off at 10^−21^. Visualization of the gene homologue numbers identified was performed using the corrplot package in R [83].

## 3. Results

### 3.1. Growth under Oligotrophic Conditions and at 37 °C

All the strains of the seven species included in this study (Table 1) grew under oligotrophic conditions, as 100-fold diluted defined medium and on agar without any added carbon or nitrogen sources (Table 2).

Growth at 37 °C on all the media tested (including oligotrophic medium) was seen for *E. dermatitidis* and *A. melanogenum*. Among the strains used for biocontrol *A. pullulans* and *D. hansenii* did not grow at 37 °C, whereas *R. mucilaginosa*, *M. guillermondii* and *M. fructicola* did grow at 37 °C, on both oligotrophic and standard regular media.

### 3.2. Biofilm Quantification

Quantification of biofilm formation determined with the crystal violet assay (Figure 1) showed that all four strains of *A. melanogenum* and *E. dermatitidis* were good biofilm producers (estimated according to biofilm-bound crystal violet absorbance at 590 nm), whereas biofilm formation of *A. pullulans*, *D. hansenii*, *R. mucilaginosa*, *M. fructicola* and *M. guillermondii* was weaker.

### 3.3. Tolerance to H_2_O_2_ Oxidative Stress

All the strains showed high tolerance to oxidative stress induced by 30 min to 60 min exposure to 20 mM H_2_O_2_. The spot assays showed almost no changes between the control conditions and the cell suspensions treated with H_2_O_2_ (Figure 2A).

In silico prediction of catalases, SODs and peroxiredoxins in the proteomes showed that the representative proteomes of all the investigated species had numerous homologues of these enzymes (Figure 2B). *A. melanogenum* and *A. pullulans* encoded the highest number of catalases (five, six respectively), and relatively low number of SODs (three, five) and peroxiredoxins (four each). *M. fructicola* had the highest number of SODs and peroxiredonis (seven of each) but has only one catalase homologue. A similar situation was seen for *D. hansenii* and *M. guilliermondii*, whereas *R. mucilaginosa* and *E. dermatitidis* had approximately equal numbers of all three H_2_O_2_-scavenging enzyme genes.

### 3.4. Siderophore Production

The strains included in this study showed variable patterns of siderophore production on the CAS agar (Figure 3A). Most of these strains produced siderophores at 24 °C, as detected by discoloration around the colonies of the otherwise blue medium. The best producers of siderophores at 24 °C were *M. fructicola* EXF-6812 and *M. guillermondii* EXF-1496. At 37 °C, only *E. dermatitidis*, *M. fructicola* and *M. guilliermondii* produced siderophores.

The prediction of the NRPSs in the representative proteomes of the species included in this study uncovered that *A. pullulans* had the highest number (19) of NRPSs, following by *A. melanogenum* (11) and *E. dermatitidis* (10) (Figure 3B). The rest of these species had one or two predicted NRPSs. The phylogenetic tree showed that adenylation domains of two NRPS involved in biosynthesis of siderophores, SidD (triacetylfusarinine) and SidC (ferricrocin), were only seen for the predicted proteomes of *A. pullulans*, *A. melanogenum* and *E. dermatitidis*.

### 3.5. Enzymatic Activities

The strains included in this study were analyzed for degradation of various plant (i.e., starch, pectin, cellulose, xylanose, aesculin), fungal (i.e., chitin) and animal-related substrates (i.e., casein, gelatine, keratin), and other compounds (i.e., fatty acids, urea) (Table 3).

All these strains showed esterase activity on fatty acids, and most of the strains (except *M. fructicola*, *M. guilliermondii*) also showed high cellulolytic activity. *E. dermatitidis* also showed xylanase activity and *A. melanogenum* and *A. pullulans* also pectinase activity. Chitinase activity was detected for *A. melanogenum*, *A. pullulans* and *M. guilliermondii*, and for one strain of *D. hansenii* and *M. fructicola*. Casein and gelatine were hydrolyzed by *A. melanogenum* (four, three strains, respectively), *A. pullulans* and one strain of *M. fructicola* (only casein) and *D. hansenii* (only gelatine). None of the tested strains were degraded keratin under these conditions. The urease test was positive for all the strains of *E. dermatitidis*, *A. melanogenum*, *A. pullulans* and *R. mucilaginosa*.

The enzymatic activities determined in the agar plate tests were in line with the prediction of urease homologues and CAZy in the computational analysis, which revealed a plethora of secreted CAZy proteins in the predicted proteomes (Figure 4). Among these, *A. pullulans* and *A. melanogenum* stand out as the producers of the widest variety of CAZy groups. Mid-level diversity of production of CAZy families was seen for *E. dermatitidis*, with lower numbers for *M. guilliermondii*, *R. mucilaginosa*, *D. hansenii* and *M. fructicola*. There were families of glycoside hydrolases in all the species, which were seen as GH1, GH3, GH5, GH16 and GH17, and others that were unique to individual species (Figure 4). *A. pullulans* and *A. melanogenum* had the highest numbers of CAZy family homologues involved in plant biomass degradation, as seen in particular by the lignocelluloses: cellulases of the GH1, GH3, GH5 and GH45 families; and xylanases of the GH3, GH10 and GH11 families. The polysaccharide lyases mainly degrade glycosaminoglycans and pectin, and these were exclusively found for *A. pullulans* and *A. melanogenum*, as PL1, PL3, PL4 and PL26. Other CAZy families involved in pectin degradation, such as CE8, GH28 and GH78, were seen for the two *Aureobasidium* species. Cutinases (i.e., family CE5) are involved in cleavage of ester bonds of waxy cutin, and these were also identified in *A. pullulans* and *A. melanogenum*.

The chitin component of the fungal cell wall can be degraded by *A. pullulans* and *A. melanogenum*, which had enzymes with β-1,3-glucanase activities of the families GH55, GH64 and GH81 [11,41]. Chitin is a known substrate for enzymes also of the GH18, GH20, and GH76 families, which were seen for most of the species included here.

### 3.6. Assimilation of Hydrocarbons

Among the three substrates of complex hydrocarbons examined here, most of the strains grew better with the aromatic hydrocarbon toluene vapors compared to the control without carbon, which indicated that these strains can assimilate toluene, except for three of the species: *D. hansenii*, *M. fructicola* and *M. guilliermondii*. A smaller proportion of the strains assimilated the liquid sources of alkane hydrocarbons, which included *n*-hexadecane and mineral oil (i.e., mixture of alkanes, cycloalkanes). Only some of the individual strains of *E. dermatitidis* and *M. guillermondii* assimilated both the *n*-hexadecane and mineral oil (Table 4).

The search for the enzymes involved in the degradation of toluene showed that *E. dermatitidis*, *A. melanogenum* and *A. pullulans* have the most complete toluene degradation pathways, with the highest numbers of predicted homologous of the individual enzymes involved (Figure 5).

## 4. Discussion

Polyextremotolerant fungi show substantial stress tolerance and have great adaptability to lowered water activity, high concentrations of various solutes, variable pHs, high and low temperatures, and low availability of nutrients [94,95]. Many polyextremotolerant fungi are considered to be ubiquitous, and they are frequently found on plant surfaces. Consequently, several of these have been proposed as and also used as biocontrol agents in agriculture [40,52,96,97]. Their ability to tolerate high stress conditions and to adapt to new habitats appears to allow them to colonize certain man-made indoor habitats that are characterized by harsh conditions [6,13,20,44]. Some have also been reported as opportunistic human pathogens [94].

What distinguishes the few fungi that have the potential to become pathogenic from the rest, as the large majority? The ability to survive and adapt to various abiotic stresses appears to be at least a part of the answer. An association between stress tolerance and virulence is indicated through the close evolutionary relatedness of fungal extremotolerance and opportunism [4,98]. Indeed, most infectious fungi are of environmental origin, where they are frequently exposed to high stress conditions similar to those encountered in human hosts. This ‘environmental training’ results in preadaptation (exaptation), which allows them to become established in the environment, and also promotes their survival within the host [99,100]. Several genes involved in stress responses are differentially expressed during fungal infections [101] and vice versa, it has been suggested that several virulence factors have primarily evolved in response to extreme environments [102,103]. A comparison of adaptations linked to environmental stress and virulence reveals a long list of traits that are useful both for growth in extreme environments (and are thus desirable for biocontrol) and for invading a mammalian body.

The relatively good resistance of mammals to fungal diseases is believed to be a combination of high body temperature and adaptive immunity [104]. The ability of a fungus to thrive at 37 °C and above (i.e., thermotolerance) is crucial for successful growth in these hosts [105] and is typical of (opportunistic) pathogenic fungi, including *E. dermatitidis* and *A. melanogenum*. The facilitated thermal adaptation of fungi, and thus the possible emergence of new pathogens, is proposed to be due to global warming and anthropogenic high-temperature habitats in our domestic environments [94]. In the present study, however, growth at 37 °C was not limited to the two pathogenic species. Instead, this was also observed in species with potential for use in biocontrol, although not for *A. pullulans* and *D. hansenii*. This trait should be viewed as the most important risk factor when assessing the safety of potential biocontrol agents.

The ability to grow under oligotrophic conditions that are characterized by low availability of nutrients is advantageous in the mammalian host, where nutrient limitation is an important defense mechanism, as well as for the competition with other microorganisms on plant surfaces. Generalistic polyextremotolerant species use various mechanisms to overcome starvation. This is clearly demonstrated in the case of iron, which is a limiting factor on plants as well as in animal hosts. Iron is essential as a cofactor for various enzymes, oxygen carriers, and electron-transfer systems that are involved in respiration, and it is also required for DNA replication [106]. During an infection in mammals, phagocytes restrict fungal growth by releasing mediators that sequester iron [107]. Under iron-limited conditions, it is of central importance to the survival of a microbe to produce high-affinity iron-chelating compounds; i.e., the siderophores [108]. In various fungi, these have roles also in virulence, fungal–host interactions, and resistance to oxidative stress [109].

All the fungi included in the present study were oligotrophic, and many of them also grew at 37 °C and under low nutrient concentrations. Our screening of siderophore production revealed that *E. dermatitidis*, *M. guilliermondii* and *M. fructicola* (although only one strain) produced siderophores at 37 °C, which is a trait of great importance in virulence, as this would hypothetically allow iron to be obtained from transferrins, ferritin, hemoglobin, and other iron-containing proteins of their host. In the present study, *A. pullulans*, *A. melanogenum* and *D. hansenii* (one strain) produced siderophores at ambient temperature but not at the human body temperature of 37 °C. Interestingly, for *A. pullulans* siderophores have been reported to act as antimicrobials, possibly through a role in the biocontrol abilities of this species [110,111].

A crucial step in the colonization of a surface of any kind, including plant or animal cells, is adhesion of the microbial cells to the surface and formation of a biofilm—a consortia of microorganisms that are adhered to the surface and embedded in extracellular polymeric substances. The formation of a biofilm is one of the most important virulence factors of a microorganism, as mature biofilms provide increased resilience to any attack by the host immune defenses, as well as to any administered antimicrobial agents [112,113]. Our data here on biofilm formation show that the strains of the opportunistic pathogens *A. melanogenum* and *E. dermatitidis* are more potent biofilm producers compared to the strains of the potential biocontrol agents. This is in line with previous observations that clinical strains of *R. mucilaginosa* are better biofilm producers compared to environmental strains [114]. On the other hand, formation of a biofilm is of benefit when a microorganism is used as a biocontrol agent, as this brings several advantages over planktonic lifestyle: higher stress tolerance, antagonism by niche exclusion due to intensive competition for space and nutrients, and antibiosis via production of antimicrobials (reviewed in [115]). It has been shown that *A. pullulans* can live on plant surfaces in the form of biofilms [116,117] and that this biofilm formation improves the biocontrol activity against sour rot on citruses [116]. The previously reported biofilm production for *D. hansenii* [118], albeit not in the context of biocontrol, was also seen here, but not for *M. fructicola* strains.

Cells are protected not only by the extracellular polysaccharides that contribute to biofilm formation, but also by melanization of the cell walls. Melanin shields fungal cells by acting as a scavenger of reactive oxygen species, and it can increase resistance to lysis and phagocytosis, and to clinically used antifungal agents [119]; it is important for tolerance to high salinity [120]. Therefore, melanization of biocontrol agents should be regarded as one of the risk factors that can promote opportunistic infections in humans. For the fungi studied here, the pathogenic *E. dermatitidis* and *A. melanogenum* are heavily melanized, whereas melanization of *A. pullulans* appears to be strain specific [121].

Tolerance to oxidative stress is tightly linked to the lifestyle of polyextremotolerant fungi. Oxidative stress can be triggered by various abiotic stressors, such as light, high salinity, extreme temperatures, starvation, and mechanical damage [89,122]. It also has roles in virulence: the oxidative burst is a crucial tool that is used against fungi as part of an ancient animal and plant immune response [107]. The ability to tolerate such oxidative bursts should therefore help a fungus to defend itself during an infection. Indeed, it has been shown that the antioxidant pathways of fungi are important for their survival against attacks by neutrophils [123]. All the yeasts in this study showed high tolerance to oxidative stress induced by 20 mM H_2_O_2_, and numerous homologues of the three major enzymes involved in cellular oxidative stress responses (i.e., catalases, SODs, peroxiredoxins) were predicted in their proteomes. Interestingly, the results of the in silico analyses here suggest that if a species has a low number of catalases, it has a higher number of the other two enzymes—SODs and peroxiredoxins. However, to confirm this trend seen here, additional, and more extensive analyses are required.

Survival on plants or in animal hosts is also linked to the production of a wide repertoire of extracellular metabolites that have important roles in interactions between fungi and their environments, i.e., secretome. The plethora of plant biomass degrading CAZys, pectat lyases and cutinases are characteristic of plant pathogenic fungi [124,125], although not every fungus with such an enzymatic profile appears to be a plant pathogen. *A. pullulans* and *A. melanogenum* show high activities for enzymes involved in plant biomass degradation (e.g., cellulase, β-glucosidase, xylanase, amylases, pectinase) and also in fungal cell-wall degradation (e.g., chitinase) (see also [5]), even though neither of these species are known to cause any plant disease. The secretome is important for the biocontrol potential of a species. It is particularly desirable for a biocontrol yeast to produce chitinases [126], so as to be able to lyse the cell walls of their competitive fungi, as seen for *A. pullulans* and *M. guillermondii*.

Urease activity is directly linked to animal pathogenesis in bacteria and fungi [127,128]. The suggested mechanism of virulence is the pH changes due to urease activity, which reduces acidification and maturation of phagolysosomes in phagocytic cells, thus impairing pathogen killing and antigen presentation [128]. In the present study, *M. fructicola*, *M. guilliermondii* and *D. hansenii* showed no urease activity, whereas for the rest of the fungi urease activities were confirmed. According to the role of urease in pathogenesis of *Cryptococcus neoformans* [127] it can be speculated that urease activity can also be considered as one of the important risk factors of biocontrol strains.

Some yeasts, such as *E. dermatitidis*, *Aureobasidium* spp. [129], and *R. mucilaginosa* [130] are frequently encountered in hydrocarbon-polluted environments [21,131,132]. A physiological connection between hydrocarbon assimilation and certain patterns of neural infection was suggested by Prenafeta- Boldú et al. (2006) [102]. We assessed the assimilation of the complex hydrocarbon toluene by the selected fungi here when used as the sole source of carbon and energy, in terms of a trait linked to virulence. Most of these strains used toluene as a carbon source, and few also used *n*-hexadecane and mineral oil. In agreement with this, the computational analysis of the proteins involved in toluene degradation showed that the black yeasts *E. dermatitidis*, *A. pullulans* and *A. melanogenum*, have almost complete pathways for degradation of toluene, which even included additional homologues of several of these enzymes. This was reflected in their good growth on toluene as the sole carbon source. Nevertheless, the assimilation of hydrocarbons examined here provide inconclusive results in terms of whether this trait can be considered as an indication of a pathogenic potential.

To sum up, the importance of fungal human pathogens has been considered to be relatively low in the past, especially when compared to bacteria and viruses. However, the growing and ageing human population, increasing numbers of immunocompromised individuals, and rapid environmental changes favor the adaptable polyextremotolerant species in our surroundings. Nowadays, alternative means of pest management that can replace the use of chemical pesticides are highly sought after for reduction of negative environmental impacts on agricultural practice. Among the alternatives, biocontrol agents based on yeasts are gaining importance, due to their effectiveness against fungal and bacterial plant pathogens, together with their robust stress tolerance (polyextremotolerance). However, several traits desirable for application to biocontrol overlap with traits that have a significant role in pathogenesis in humans, such as oligotrophism, siderophore production, biofilm formation and tolerance to oxidative stress (Figure 6). Therefore, special caution should be addressed to individual strains, and more precisely, to their ability to grow and produce siderophores at 37 °C, their urease and proteolytic activities, and even the melanization of their cell walls. As fungal strains with (several of) these traits might cause diseases in humans under specific circumstances, these need to be carefully examined when assessing the safety of potential biocontrol agents.

## Figures and Tables

**Figure 1 genes-10-00042-f001:**
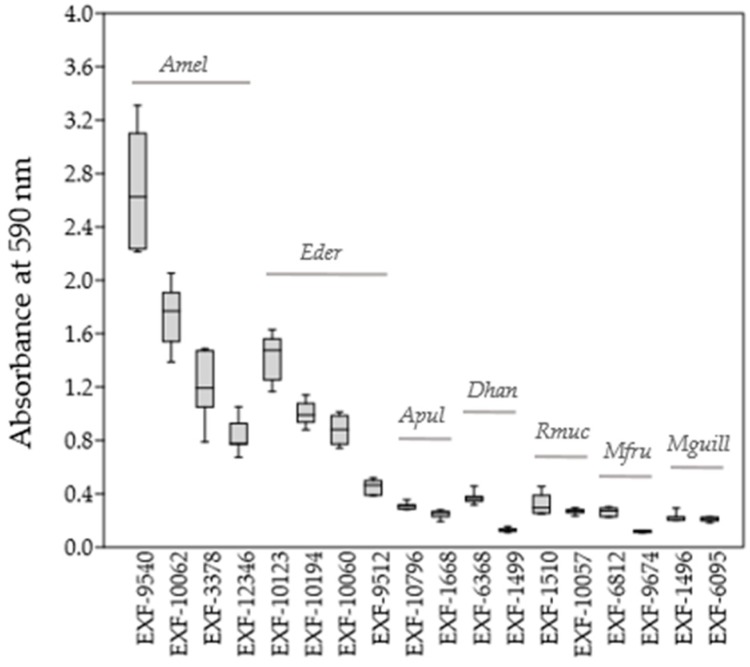
Quantification of biofilm formation by the selected yeast strains using the crystal violet assay.

**Figure 2 genes-10-00042-f002:**
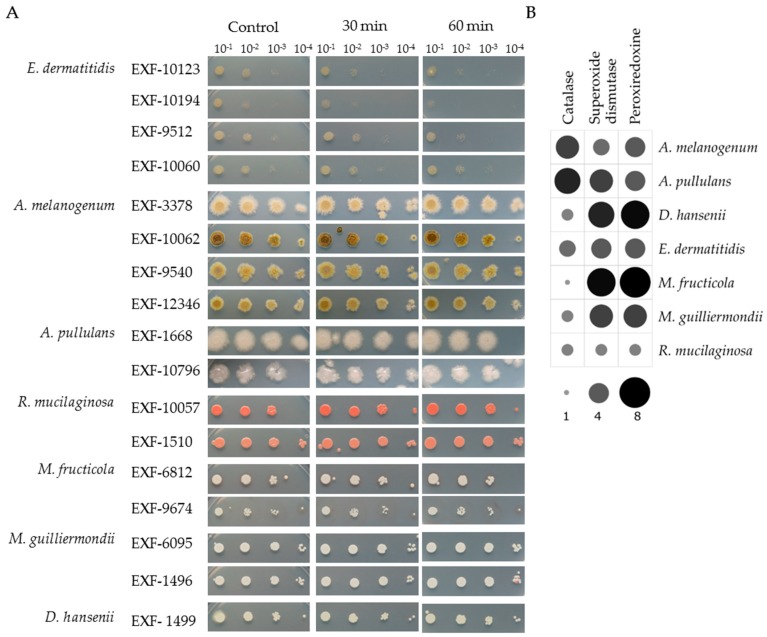
(**A**). Spot assays for the selected yeast strains under oxidative stress due to 30 min and 60 min exposure to 20 mM H_2_O_2_compared to the control. (**B**). Predicted catalases, superoxide dismutases and peroxiredoxins in the representative proteomes. The size and color intensity of dots corresponds to the number of homologues.

**Figure 3 genes-10-00042-f003:**
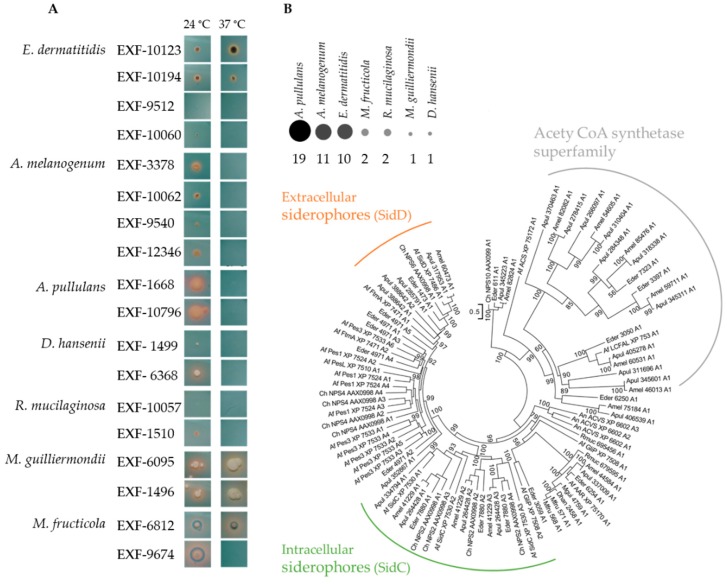
(**A**). Production of siderophores by the selected yeast strains on chrome azurol S agar (CAS) (yellow to orange halos around colonies indicate siderophore production). (**B**). Prediction of the non-ribosomal peptide synthetases (NRPSs) in the proteomes of the selected yeast strains, together with the phylogenetic tree of their adenylation domains. The size and color intensity of dots corresponds to the number of homologues.

**Figure 4 genes-10-00042-f004:**
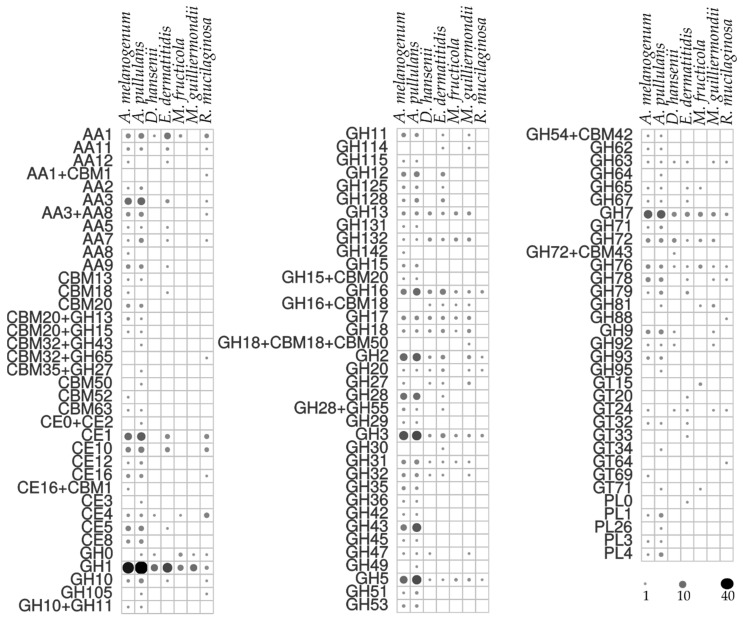
Predicted numbers of carbohydrate active enzymes (CAZy) in the proteomes of the selected yeast species, according to the dbCAN server. The size and color intensity of dots corresponds to the number of homologues.

**Figure 5 genes-10-00042-f005:**
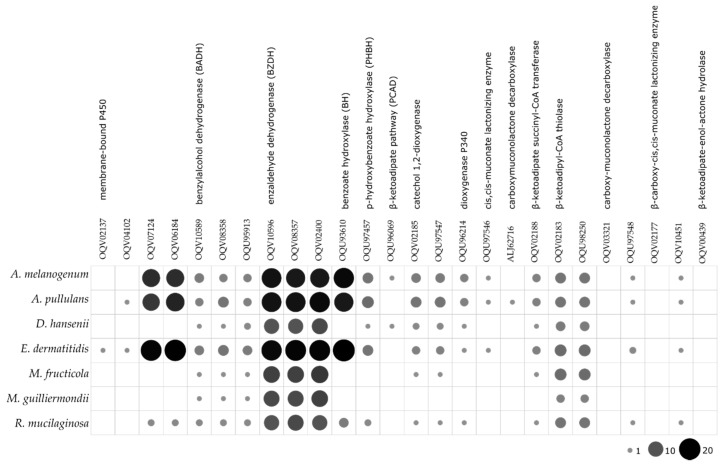
Predicted numbers of enzymes involved in the toluene degradation pathway in the proteomes of the selected yeast species. The size and color intensity of dots corresponds to the number of homologues.

**Figure 6 genes-10-00042-f006:**
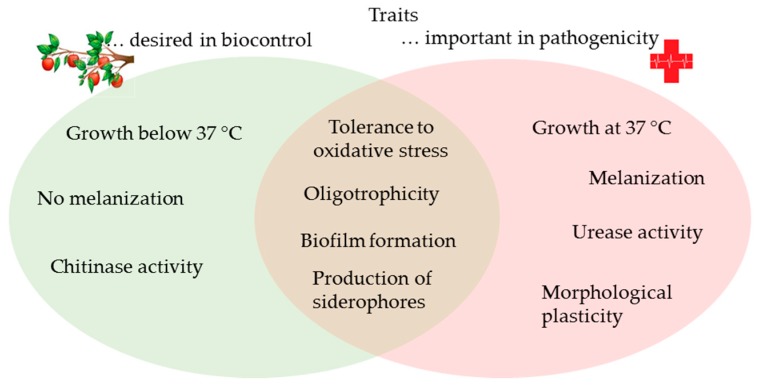
Overlap between the traits involved in pathogenesis and that are desirable for antagonism in biocontrol.

**Table 1 genes-10-00042-t001:** Selected yeast strains used in this study.

Species	Accession Number	Source of Isolation
*Exophiala dermatitidis*	EXF-10123 (CBS 525.76)	Human
	EXF-10194	Patient with cystic fibrosis, Sweden
	EXF-9512	Dishwasher rubber, Slovenia
	EXF-10060	Tap water, Slovenia
*Aureobasidium melanogenum*	EXF-3378 (CBS 110374)	Public fountain, Thailand
	EXF-10062	Tap water, Slovenia
	EXF-9540	Dishwasher rubber, Slovenia
	EXF-12346	Endoperitoneal fluid, India
*Aureobasidium pullulans*	EXF-1668	Glacial ice, Arctic
	EXF-10796	Grape surface, Slovenia
*Debaryomyces hansenii*	EXF-1499	Glacial ice, Arctic
	EXF-6368	Car fuel, Slovenia
*Rhodotorula mucilaginosa*	EXF-10057	Tap water, Slovenia
	EXF-1510	Glacial ice, Arctic
*Metschnikowia fructicola*	EXF-6812	Grape surface, Slovenia
	EXF-9674	Dishwasher drain, Slovenia
*Meyerozyma guilliermondii*	EXF-6095	Dishwasher rubber seal, Slovenia
	EXF-1496	Glacial ice, Arctic

**Table 2 genes-10-00042-t002:** Growth of the selected yeast strains at 24 °C and 37 °C on yeast nitrogen base (YNB), 100-fold diluted YNB (1/100 YNB) and pure agar medium (Agar).

Species		24 °C			37 °C		
	Accession Number	YNB	1/100 YNB	Agar	YNB	1/100 YNB	Agar
*Exophiala dermatitidis*	EXF-10123	**+**	**+**	**+**	**+**	**+**	**+**
	EXF-10194	**+**	**+**	**+**	**+**	**+**	**+**
	EXF-9512	**+**	**+**	**+**	**+**	**+**	**+**
	EXF-10060	**+**	**+**	**+**	**+**	**+**	**+**
*Aureobasidium melanogenum*	EXF-3378	**+**	**+**	**+**	**+**	**+**	**+**
	EXF-10062	**+**	**+**	**+**	**+**	**+**	**+**
	EXF-9540	**+**	**+**	**+**	**+**	**+**	**+**
	EXF-12346	**+**	**+**	**+**	**+**	**+**	**+**
*Aureobasidium pullulans*	EXF-1668	**+**	**+**	**+**	**-**	**-**	**-**
	EXF-10796	**+**	**+**	**+**	**-**	**-**	**-**
*Debaryomyces hansenii*	EXF-1499	**+**	**+**	**+**	**-**	**-**	**-**
	EXF-6368	**+**	**+/-**	**+/-**	**-**	**-**	**-**
*Rhodotorula mucilaginosa*	EXF-10057	**+**	**+**	**+**	**+**	**+**	**+**
	EXF-1510	**+**	**+**	**+**	**+**	**+**	**+**
*Metchnikowia fructicola*	EXF-6812	**+**	**+**	**+**	**+**	**+**	**-**
	EXF-9674	**+**	**+**	**+**	**+/-**	**+/-**	**-**
*Meyerozyma guilliermondii*	EXF-6095	**+**	**+**	**+**	**+**	**+**	**+**
	EXF-1496	**+**	**+**	**+**	**+**	**+**	**+**

+ good growth, +/- weak growth and - no growth.

**Table 3 genes-10-00042-t003:** Enzymatic activities of the selected yeast strains.

Species	Strain	Enzymatic Activity										
		Amylases (EI)	Pectinase (EI)	Cellulase (EI)	Xylanase (EI)	Chitinase	β-Glucosidase	Esterase	Caseinase	Gelatinase	Keratinase	Urease
*Exophiala dermatitidis*	**EXF-10123**	-	-	1.25 ± 0.16	1.12 ± 0.21	-	-	+/−	-	-	-	+
	**EXF-10194**	-	-	1.48 ± 0.28	+/−	-	-	+/−	-	-	-	+/−
	**EXF-9512**	-	-	1.45 ± 0.90	1.29 ± 0.14	-	-	+/−	-	-	-	+
	**EXF-10060**	-	-	1.46 ± 0.11	1.22 ± 0.14	-	-	+	-	-	-	+
*Aureobasidium melanogenum*	**EXF-3378**	-	+/−	1.42 ± 0.27	-	+	+	+	+	+	-	+
	**EXF-10062**	1.30 ± 0.28	1.35 ± 0.30	1.31 ± 0.14	-	+	+	+	+	+	-	+
	**EXF-9540**	-	1.23 ± 0.10	1.3 2± 0.2	+/−	+	+	+	+	-	-	+
	**EXF-12346**	-	1.23 ± 0.20	1.42 ± 0.33	1.18 ± 0.27	+	+	+	+	+	-	+
*Aureobasidium pullulans*	**EXF-1668**	1.30 ± 0.08	1.38 ± 0.18	1.86 ± 0.67	-	+	+	+	+	+	-	+
	**EXF-10796**	1.19 ± 0.12	1.23 ± 0.06	1.56 ± 0.51	-	+	+	+	+	+	-	+
*Debaryomyces hansenii*	**EXF-1499**	-	1.33 ± 0.12	1.20 ± 0.00	1.25 ± 0.00	-	-	+/−	-	+	-	-
	**EXF-6368**	-	-	1.19 ± 0.17	-	+	-	+	-	-	-	-
*Rhodotorula mucilaginosa*	**EXF-10057**	-	-	1.31 ± 0.22	1.60 ± 0.44	-	-	+	-	-	-	+
	**EXF-1510**	+/−	-	1.29 ± 0.47	+/−	-	-	+	-	-	-	+
*Metschnikowia fructicola*	**EXF-6812**	+/−	-	-	+/−	-	-	+	-	-	-	-
	**EXF-9674**	-	-	-	-	+	-	+	+	-	-	-
*Meyerozyma guilliermondii*	**EXF-6095**	-	-	-	-	+	-	+	-	-	-	-
	**EXF-1496**	-	-	+/−	-	+	-	+	-	-	-	-

EI, enzymatic index (numeric; mean ± SD); +, good activity; +/−, weak activity; -, no activity (descriptive).

**Table 4 genes-10-00042-t004:** Assimilation of the hydrocarbons by the selected yeast strains.

Species	Strain	Assimilation Activity		
		Toluene	*n*-Hexadecane	Mineral Oil
*Exophiala dermatitidis*	EXF-10123	+	+	+
	EXF-10194	+	-	-
	EXF-9512	+	-	-
	EXF-10060	+	-	+
*Aureobasidium melanogenum*	EXF-3378	+	-	-
	EXF-10062	+	-	-
	EXF-9540	+	-	+
	EXF-12346	+	-	+
*Aureobasidium pullulans*	EXF-1668	+	-	-
	EXF-10796	+	-	-
*Debaryomyces hansenii*	EXF-1499	-	-	-
	EXF-6368	-	+	-
*Rhodotorula mucilaginosa*	EXF-10057	+	-	-
	EXF-1510	+	-	-
*Metschnikowia fructicola*	EXF-6812	-	+	-
	EXF-9674	-	+	-
*Meyerozyma guilliermondii*	EXF-6095	-	+	+
	EXF-1496	-	+	+

+, good activity; +/-, weak activity; -, no activity.

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
