# Peer review of "Stress-Tolerant Yeasts: Opportunistic Pathogenicity Versus Biocontrol Potential"

_genes, 2019, doi:10.3390/genes10010042_

Round 1
Reviewer 1 Report
I would just like to point out some typos I have noticed while reading the manuscript.
line 115: mamamals
line 174: detrmined
line 392: polyextremotlerant.
Moreover, under the author contributions, the names of those responsible for Writing- reviewing and editing, supervision and project administrations are missing.
Author Response
Dr. Janja Zajc
National Institute of Biology
Večna pot 111
SI-1000 Ljubljana
Slovenia
We are submitting a revised version of the article entitled “Stress-tolerant yeasts: opportunistic pathogenicity versus biocontrol potential” by Janja Zajc, Cene Gostinčar, Anja Černoša and Nina Gunde-Cimerman to the journal Genes (ISSN 2073-4425).
The manuscript has been extensively revised and all the comments of the reviewers were carefully addressed. Importantly, the growth limits and/or growth ranges of the fungi described here were defined; figures were supplemented with additional information and the discussion was substantially shortened. We feel that revised version of the manuscript is improved and we would like to thank all the reviewers for their constructive comments.
Please find the response to reviewer 1 in the word document attached below.
Sincerely yours,
Ljubljana, 3. 1. 2019 Dr. Janja Zajc

Reviewer 2 Report
The manuscript entitled “Stress-tolerant yeasts: opportunistic pathogenicity versus biocontrol potential” described a series of experiments to examine which traits are shared between pathogenic and biocontrol strains. This manuscript is of potential interest for the Genes readers and will complements previous research of stress tolerant yeasts, therefore being of great importance for a broader audience. Overall this manuscript is interesting but could benefits from some edits.
Major comments:
Often times the authors mentioned that a certain organism was extremotolerant or tolerant to a specific condition, such as thermotolerant. In addition to providing references for these statements the authors should include in parenthesis the growth ranges, for example low and upper temperatures. This way it is easier to know what the authors means with “tolerant” and will avoid ambiguities with this term. For example, in line 68 the authors said “its pronounced thermotolerance” but I am not sure what this really means.
Lines 110-116: It is clear that having these traits will facilitate the organism to be more adapted to different environmental conditions and therefore expanding their niches. However, it is not clear how these traits will define their roles in biocontrol or/and in virulence. To clarify this, the authors should briefly explain how these different traits are desirable for one or another.
Why the biofilm production, enzymatic activities, tolerance to oxidative stress, and the assimilation of hydrocarbon assays where measured at 24C and not at 37C or both? Please explain.
Lines 521-536: The authors did a great job summarizing why these traits could be desirable for biocontrol or could be of risk for human health. However, the authors should describe why and/or how organisms that have traits that are shared with pathogenic strains (i.e grow and production of siderophores at 37 °C, urease and proteolytic activities, etc) might cause diseases in the future (since these are not currently pathogenic). For example, is there any report about horizontal gene transfer that could explain the transfer of virulence factors between pathogenic and non-pathogenic strains?
Minor comments:
Lines 45-46: Please check with the journal policies if the taxonomy names above the genus level should be italicized or not.
Line 48: add space between be and safe (to be safe not to besafe)
Line 116: I would remove “extreme” since the observation that an organism is thermotolerant because it can grow at 45C dos not necessarily mean that the environment where is present is an extreme environment. In fact, 45C is within the growth range of mesophilic microorganisms and by no means is considered an “extreme temperature”.
Line 128: Remove “The” from table heading
Table 1: Please indicate if the accession number belong to Mycosmo or the Westerdijk Fungal Biodiversity Institute.
Line 130: Remove “and at 37C”
Lines 132-133: Remove “in deionized water” because it was previously mentioned.
Line 141: Change “w/V” to “w/v”
Line 143: “590 nm” is written twice. Remove one.
Lines 216-221: Please give more information about the dilution used. For example, how much was diluted, which dilution factor was used. Why deionized water was used for the dilutions and not the culture media?
Line 233: Remove “and”
Line 244: Remove “and”
Line 248: Replace “scanning” by “screening”
Line 280: Remove “and at 37C”
Line 293: Remove “extremely”
Line 300-302: Please include more information about the control in the method section. How was the control done?
Figure 2A: I assume that the four different spots corresponded to the serial dilutions. Please include for each row the dilution that was plated (i.e. 101,102, 103, etc). 2B: More information should be included in the figure legend to explain what the color and size of the different circles mean.
Lines 316-320 and figure 3A: Please indicate in Figure 3A the row that corresponded to 24C and 37C. Why some strains grew at 24C but not at 37C? For example, all the A. melanogenum strains did not grew at 37C but from the information presented in table 2 these strains should grow at 37C. Is this because in order to growth in this CAS agar the microorganism require siderophores? If this is the case, this should be described in the text. Please explain.
Line 343: Remove “were”
Figure 4 and 5: Similar to figure 2B, more information should be included in the figure legend to explain what the color and size of the different circles mean.
Line 366-367: What do the authors mean with “carbohydrate active”? Please be more specific.
Author Response
Dr. Janja Zajc
National Institute of Biology
Večna pot 111
SI-1000 Ljubljana
Slovenia
We are submitting a revised version of the article entitled “Stress-tolerant yeasts: opportunistic pathogenicity versus biocontrol potential” by Janja Zajc, Cene Gostinčar, Anja Černoša and Nina Gunde-Cimerman to the journal Genes (ISSN 2073-4425).
The manuscript has been extensively revised and all the comments of the reviewers were carefully addressed. Importantly, the growth limits and/or growth ranges of the fungi described here were defined; figures were supplemented with additional information and the discussion was substantially shortened. We feel that revised version of the manuscript is improved and we would like to thank all the reviewers for their constructive comments.
Please find the response to reviewer 2 in the word document attached below.
Sincerely yours,
Ljubljana, 3. 1. 2019 Dr. Janja Zajc

Reviewer 3 Report
In general, the paper is well written and also deals with a very current and interesting topic.
My main request concerns the need to summarize the discussion, avoiding repeating concepts already expressed in the introduction.
Other small corrections:
page 2 -line 48: be safe
page 4 - line 143: "590 nm" repeated twice
page 5 - line 209: "follows: first"
page 15 - line 463: "was also seen here,"
page 16 - line 501: activities
Author Response
Dr. Janja Zajc
National Institute of Biology
Večna pot 111
SI-1000 Ljubljana
Slovenia
We are submitting a revised version of the article entitled “Stress-tolerant yeasts: opportunistic pathogenicity versus biocontrol potential” by Janja Zajc, Cene Gostinčar, Anja Černoša and Nina Gunde-Cimerman to the journal Genes (ISSN 2073-4425).
The manuscript has been extensively revised and all the comments of the reviewers were carefully addressed. Importantly, the growth limits and/or growth ranges of the fungi described here were defined; figures were supplemented with additional information and the discussion was substantially shortened. We feel that revised version of the manuscript is improved and we would like to thank all the reviewers for their constructive comments.
Please find the response to reviewer 3 in the word document attached below.
Sincerely yours,
Ljubljana, 3. 1. 2019 Dr. Janja Zajc
